# Learning from Rational Behavior:
# Predicting Solutions to Unknown Linear Programs

**Shahin Jabbari, Ryan Rogers, Aaron Roth, Zhiwei Steven Wu**
University of Pennsylvania
{jabbari@cis, ryrogers@sas, aaroth@cis, wuzhiwei@cis}.upenn.edu

## Abstract

We define and study the problem of predicting the solution to a linear program (LP) given only partial information about its objective and constraints. This generalizes the problem of learning to predict the purchasing behavior of a rational agent who has an unknown objective function, that has been studied under the name "Learning from Revealed Preferences". We give mistake bound learning algorithms in two settings: in the first, the *objective* of the LP is known to the learner but there is an arbitrary, fixed set of constraints which are unknown. Each example is defined by an additional known constraint and the goal of the learner is to predict the optimal solution of the LP given the union of the known and unknown constraints. This models the problem of predicting the behavior of a rational agent whose *goals* are known, but whose resources are unknown. In the second setting, the objective of the LP is unknown, and changing in a controlled way. The *constraints* of the LP may also change every day, but are known. An example is given by a set of constraints and partial information about the objective, and the task of the learner is again to predict the optimal solution of the partially known LP.

## 1 Introduction

We initiate the systematic study of a general class of multi-dimensional prediction problems, where the learner wishes to predict the solution to an unknown linear program (LP), given some partial information about either the set of constraints or the objective. In the special case in which there is a single known *constraint* that is changing and the objective that is unknown and fixed, this problem has been studied under the name *learning from revealed preferences* [1, 2, 3, 16] and captures the following scenario: a buyer, with an unknown linear utility function over $d$ goods $u : \mathbb{R}^d \to \mathbb{R}$ defined as $u(\mathbf{x}) = \mathbf{c} \cdot \mathbf{x}$ faces a purchasing decision every day. On day $t$, she observes a set of prices $\mathbf{p}^t \in \mathbb{R}^d_{\geq 0}$ and buys the bundle of goods that maximizes her unknown utility, subject to a *budget* $b$:

$$\mathbf{x}^{(t)} = \operatorname*{argmax}_{\mathbf{x}} \mathbf{c} \cdot \mathbf{x} \qquad \text{such that } \mathbf{p}^t \cdot \mathbf{x} \leq b$$

In this problem, the goal of the learner is to predict the bundle that the buyer will buy, given the prices that she faces. Each example at day $t$ is specified by the vector $\mathbf{p}^t \in \mathbb{R}^d_{\geq 0}$ (which fixes the constraint), and the goal is to accurately predict the purchased bundle $\mathbf{x}^{(t)} \in [0,1]^d$ that is the result of optimizing the unknown linear objective.

It is also natural to consider the class of problems in which the goal is to predict the outcome to a LP broadly e.g. suppose the objective $\mathbf{c} \cdot \mathbf{x}$ is known but there is an *unknown* set of constraints $A\mathbf{x} \leq \mathbf{b}$. An *instance* is again specified by a changing known constraint $(\mathbf{p}^t, b^t)$ and the goal is to predict:

$$\mathbf{x}^{(t)} = \operatorname*{argmax}_{\mathbf{x}} \mathbf{c} \cdot \mathbf{x} \qquad \text{such that } A\mathbf{x} \leq \mathbf{b} \quad \text{and } \mathbf{p}^t \cdot \mathbf{x} \leq b^t. \tag{1}$$

This models the problem of predicting the behavior of an agent whose *goals* are known, but whose resource constraints are unknown.

Another natural generalization is the problem in which the objective is unknown, and may vary in a specified way across examples, and in which there may also be multiple arbitrary known constraints which vary across examples. Specifically, suppose that there are $n$ distinct, unknown linear objective functions $\mathbf{v}^1, \ldots, \mathbf{v}^n$. An *instance* on day $t$ is specified by a subset of the unknown objective functions, $S^t \subseteq [n] := \{1, \ldots, n\}$ and a convex feasible region $\mathcal{P}^t$, and the goal is to predict:

$$\mathbf{x}^{(t)} = \operatorname*{argmax}_{\mathbf{x}} \sum_{i \in S^t} \mathbf{v}^i \cdot \mathbf{x} \qquad \text{such that } \mathbf{x} \in \mathcal{P}^t. \tag{2}$$

When the changing feasible regions $\mathcal{P}^t$ correspond simply to varying prices as in the revealed preferences problem, this models a setting in which at different times, purchasing decisions are made by different members of an organization, with heterogeneous preferences — but are still bound by an organization-wide budget. The learner's problem is, given the subset of decision makers and the prices at day $t$, to predict which bundle they will purchase. This generalizes some of the preference learning problems recently studied by Blum et al [6]. Of course, in this generality, we may also consider a richer set of changing constraints which represent things beyond prices and budgets.

In all of the settings we study, the problem can be viewed as the task of predicting the behavior of a rational decision maker, who always chooses the action that maximizes her objective function subject to a set of constraints. Some part of her optimization problem is unknown, and the goal is to learn, through observing her behavior, that unknown part of her optimization problem sufficiently so that we may reliably predict her future actions.

## 1.1 Our Results

We study both variants of the problem (see below) in the strong *mistake bound* model of learning [13]. In this model, the learner encounters an arbitrary adversarially chosen sequence of examples online and must make a prediction for the optimal solution in each example before seeing future examples. Whenever the learner's prediction is incorrect, the learner encounters a *mistake*, and the goal is to prove an upper bound on the number of mistakes the learner can make, in the worst case over the sequence of examples. Mistake bound learnability is stronger than (and implies) PAC learnability [15].

**Known Objective and Unknown Constraints**   We first study this problem under the assumption that there is a uniform upper bound on the number of bits of precision used to specify the constraint defining each example. In this case, we show that there is a learning algorithm with both running time and mistake bound linear in the number of edges of the polytope formed by the unknown constraint matrix $A\mathbf{x} \leq b$. We note that this is always polynomial in the dimension $d$ when the number of unknown constraints is at most $d + O(1)$. (In the supplementary material, we show that by allowing the learner to run in time exponential in $d$, we can give a mistake bound that is always linear in the dimension and the number of rows of $A$, but we leave as an open question whether or not this mistake bound can be achieved by an efficient algorithm.) We then show that our bounded precision assumption is necessary — i.e. we show that when the precision to which constraints are specified need not be uniformly upper bounded, then no algorithm for this problem in dimension $d \geq 3$ can have a finite mistake bound.

This lower bound motivates us to study a PAC style variant of the problem, where the examples are not chosen in an adversarial manner, but instead are drawn independently at random from an arbitrary unknown distribution. In this setting, we show that even if the constraints can be specified to arbitrary (even infinite) precision, there is a learner that requires sample complexity only linear in the number of edges of the unknown constraint polytope. This learner can be implemented efficiently when the constraints are specified with finite precision.

**Known Constraints and Unknown Objective**   For the variant of the problem in which the objective is unknown and changing and the constraints are known but changing, we give an algorithm that has a mistake bound and running time polynomial in the dimension $d$. Our algorithm uses the Ellipsoid algorithm to learn the coefficients of the unknown objective by implementing a separation oracle that generates separating hyperplanes given examples on which our algorithm made a mistake.

We leave the study of either of our problems under natural relaxations (e.g. under a less demanding loss function) and whether it is possible to substantially improve our results in these relaxations as an interesting open problem.

## 1.2 Related Work

Beigman and Vohra [3] were the first to study *revealed preference problems* (RPP) as a learning problems and to relate them to multi-dimensional classification. They derived sample complexity bounds for such problems by computing the fat shattering dimension of the class of target utility functions, and showed that the set of Lipschitz-continuous valuation functions had finite fat-shattering dimension. Zadimoghaddam and Roth [16] gave efficient algorithms with polynomial sample complexity for PAC learning of the RPP over the class of linear (and piecewise linear) utility functions. Balcan et al. [2] showed a connection between RPP and the structured prediction problem of learning $d$-dimensional linear classes [7, 8, 12], and use an efficient variant of the compression techniques given by Daniely and Shalev-Shwartz [9] to give efficient PAC algorithms with optimal sample complexity for various classes of economically meaningful utility functions. Amin et al. [1] study the RPP for linear valuation functions in the mistake bound model, and in the query model in which the learner gets to set prices and wishes to maximize profit. Roth et al. [14] also study the query model of learning and give results for strongly concave objective functions, leveraging an algorithm of Belloni et al. [4] for bandit convex optimization with adversarial noise.

All of the works above focus on the setting of predicting the optimizer of a fixed unknown objective function, together with a single known, changing constraint representing prices. This is the primary point of departure for our work — we give algorithms for the more general settings of predicting the optimizer of a LP when there may be many unknown constraints, or when the unknown objective function is changing. Finally, the literature on *preference learning* (see e.g. [10]) has similar goals, but is technically quite distinct: the canonical problem in preference learning is to learn a *ranking* on distinct elements. In contrast, the problem we consider here is to predict the outcome of a continuous optimization problem as a function of varying constraints.

## 2 Model and Preliminaries

We first formally define the geometric notions used throughout this paper. A *hyperplane* and a *halfspace* in $\mathbb{R}^d$ are the set of points satisfying the linear equation $a_1 x_1 + \ldots a_d x_d = b$ and the linear inequality $a_1 x_1 + \ldots + a_d x_d \leq b$ for a set of $a_i$s respectively, assuming that not all $a_i$'s are simultaneously zero. A set of *hyperplanes* are *linearly independent* if the normal vectors to the hyperplanes are linearly independent. A *polytope* (denoted by $\mathcal{P} \subseteq \mathbb{R}^d$) is the *bounded* intersection of finitely many halfspaces, written as $\mathcal{P} = \{\mathbf{x} \mid A\mathbf{x} \leq \mathbf{b}\}$. An *edge-space* $e$ of a polytope $\mathcal{P}$ is a one dimensional subspace that is the intersection of $d - 1$ linearly independent hyperplanes of $\mathcal{P}$, and an *edge* is the intersection between an edge-space $e$ and the polytope $\mathcal{P}$. We denote the set of edges of polytope $\mathcal{P}$ by $E_{\mathcal{P}}$. A *vertex* of $\mathcal{P}$ is a point where $d$ linearly independent hyperplanes of $\mathcal{P}$ intersect. Equivalently, $\mathcal{P}$ can be written as the *convex hull* of its vertices $V$ denoted by $\mathrm{Conv}(V)$. Finally, we define a set of points to be *collinear* if there exists a line that contains all the points in the set.

We study an online prediction problem with the goal of predicting the optimal solution of a changing LP whose parameters are only partially known. Formally, in each day $t = 1, 2, \ldots$ an adversary chooses a LP specified by a polytope $\mathcal{P}^{(t)}$ (a set of linear inequalities) and coefficients $\mathbf{c}^{(t)} \in \mathbb{R}^d$ of the linear objective function. The learner's goal is to predict the solution $\mathbf{x}^{(t)}$ where $\mathbf{x}^{(t)} = \mathrm{argmax}_{\mathbf{x} \in \mathcal{P}^{(t)}} \mathbf{c}^{(t)} \cdot \mathbf{x}$. After making the prediction $\hat{\mathbf{x}}^{(t)}$, the learner observes the optimal $\mathbf{x}^{(t)}$ and learns whether she has made a mistake ($\hat{\mathbf{x}}^{(t)} \neq \mathbf{x}^{(t)}$). The mistake bound is defined as follows.

**Definition 1.** Given a LP with feasible polytope $\mathcal{P}$ and objective function $\mathbf{c}$, let $\sigma^{(t)}$ denote the parameters of the LP that are revealed to the learner on day $t$. A learning algorithm $\mathcal{A}$ takes as input the sequence $\{\sigma^{(t)}\}_t$, the known parameters of an adaptively chosen sequence $\{(\mathcal{P}^{(t)}, \mathbf{c}^{(t)})\}_t$ of LPs and outputs a sequence of predictions $\{\hat{\mathbf{x}}^{(t)}\}_t$. We say that $\mathcal{A}$ has mistake bound $M$ if $\max_{\{(\mathcal{P}^{(t)}, \mathbf{c}^{(t)})\}_t} \left\{ \Sigma_{t=1}^{\infty} \mathbf{1} \left[ \hat{\mathbf{x}}^{(t)} \neq \mathbf{x}^{(t)} \right] \right\} \leq M$, where $\mathbf{x}^{(t)} = \mathrm{argmax}_{\mathbf{x} \in \mathcal{P}^{(t)}} \mathbf{c}^{(t)} \cdot \mathbf{x}$ on day $t$.

We consider two different instances of the problem described above. First, in Section 3, we study the problem given in (1) in which $\mathbf{c}^{(t)} = \mathbf{c}$ is fixed and known to the learner but the polytope $\mathcal{P}^{(t)} =$

$\mathcal{P} \cap \mathcal{N}^{(t)}$ consists of an unknown fixed polytope $\mathcal{P}$ and a new constraint $\mathcal{N}^{(t)} = \{\mathbf{x} \mid \mathbf{p}^{(t)} \cdot \mathbf{x} \leq b^{(t)}\}$ which is revealed to the learner on day $t$ i.e. $\sigma^{(t)} = (\mathcal{N}^{(t)}, \mathbf{c})$. We refer to this as the *Known Objective* problem. Then, in Section 4, we study the problem in which the polytope $\mathcal{P}^{(t)}$ is changing and known but the objective function $\mathbf{c}^{(t)} = \sum_{i \in S^{(t)}} \mathbf{v}^i$ is unknown and changing as in (2) where the set $S^{(t)}$ is known i.e. $\sigma^{(t)} = (\mathcal{P}^{(t)}, S^{(t)})$. We refer to this as the *Known Constraints* problem.

In order for our prediction problem to be well defined, we make Assumption 1 about the observed solution $\mathbf{x}^{(t)}$ in each day. Assumption 1 guarantees that each solution is on a vertex of $\mathcal{P}^{(t)}$.

**Assumption 1.** *The optimal solution to the LP:* $\max_{\mathbf{x} \in \mathcal{P}^{(t)}} \mathbf{c}^{(t)} \cdot \mathbf{x}$ *is unique for all* $t$.

## 3 The Known Objective Problem

In this section, we focus on the *Known Objective Problem* where the coefficients of the objective function $\mathbf{c}$ are fixed and known to the learner but the feasible region $\mathcal{P}^{(t)}$ on day $t$ is unknown and changing. In particular, $\mathcal{P}^{(t)}$ is the intersection of a fixed and unknown polytope $\mathcal{P} = \{\mathbf{x} \mid A\mathbf{x} \leq \mathbf{b}, A \subseteq \mathbb{R}^{m \times d}\}$ and a known halfspace $\mathcal{N}^{(t)} = \{\mathbf{x} \mid \mathbf{p}^{(t)} \cdot \mathbf{x} \leq b^{(t)}\}$ i.e. $\mathcal{P}^{(t)} = \mathcal{P} \cap \mathcal{N}^{(t)}$.

Throughout this section we make the following assumptions. First, we assume w.l.o.g. (up to scaling) that the points in $\mathcal{P}$ have $\ell_\infty$-norm bounded by 1.

**Assumption 2.** *The unknown polytope $\mathcal{P}$ lies inside the unit $\ell_\infty$-ball i.e.* $\mathcal{P} \subseteq \{\mathbf{x} \mid ||\mathbf{x}||_\infty \leq 1\}$.

We also assume that the coordinates of the vertices in $\mathcal{P}$ can be written with finite precision (this is implied if the halfspaces defining $\mathcal{P}$ can be described with finite precision). [1]

**Assumption 3.** *The coordinates of each vertex of $\mathcal{P}$ can be written with $N$ bits of precision.*

We show in Section 3.3 that Assumption 3 is necessary — without any upper bound on precision, there is no algorithm with a finite mistake bound. Next, we make some non-degeneracy assumptions on polytopes $\mathcal{P}$ and $\mathcal{P}^{(t)}$, respectively. We require these assumptions to hold on each day.

**Assumption 4.** *Any subset of $d-1$ rows of $A$ have rank $d-1$ where $A$ is the constraint matrix in* $\mathcal{P} = \{\mathbf{x} \mid A\mathbf{x} \leq \mathbf{b}\}$.

**Assumption 5.** *Each vertex of $\mathcal{P}^{(t)}$ is the intersection of exactly $d$-hyperplanes of $\mathcal{P}^{(t)}$.*

The rest of this section is organized as follows. We present `LearnEdge` for the Known Objective Problem and analyze its mistake bound in Sections 3.1 and 3.2, respectively. Then in Section 3.3, we prove the necessity of Assumption 3 to get a finite mistake bound. Finally in Section 3.4, we present the `LearnHull` in a PAC style setting where the new constraint each day is drawn i.i.d. from an unknown distribution, rather than selected adversarially.

### 3.1 `LearnEdge` Algorithm

In this section we introduce `LearnEdge` and show in Theorem 1 that the number of mistakes of `LearnEdge` depends linearly on the number of edges $E_\mathcal{P}$ and the precision parameter $N$ and only logarithmically on the dimension $d$. We defer all the missing proofs to the supplementary material.

**Theorem 1.** *The number of mistakes and per day running time of `LearnEdge` in the Known Objective Problem are $O(|E_\mathcal{P}|N\log(d))$ and $\text{poly}(m, d, |E_\mathcal{P}|)$ respectively when $A \subseteq \mathbb{R}^{m \times d}$.*

At a high level, `LearnEdge` maintains a set of *prediction information* $\mathcal{I}^{(t)}$ about the prediction history up to day $t$, and makes prediction in each day based on $\mathcal{I}^{(t)}$ and a set of *prediction rules* (**P.1** − **P.4**). After making a mistake, `LearnEdge` updates the information with a set of *update rules* (**U.1** − **U.4**).

**Prediction Information** It is natural to ask "What information is useful for prediction?" Lemma 2 establishes the importance of the set of edges $E_\mathcal{P}$ by showing that all the observed solutions will be on an element of $E_\mathcal{P}$.

**Lemma 2.** *On any day $t$, the observed solution $\mathbf{x}^{(t)}$ lies on an edge in $E_{\mathcal{P}}$.*

In the proof of Lemma 2 we also show that when $\mathbf{x}^{(t)}$ does not bind the new constraint $\mathcal{N}^{(t)}$, then $\mathbf{x}^{(t)}$ is the solution for the underlying LP: $\operatorname{argmax}_{\mathbf{x}\in\mathcal{P}} \mathbf{c} \cdot \mathbf{x}$.

**Corollary 1.** *If $\mathbf{x}^{(t)} \in \{\mathbf{x} \mid \mathbf{p}^{(t)}\mathbf{x} < b^{(t)}\}$ then $\mathbf{x}^{(t)} = \mathbf{x}^* \equiv \operatorname{argmax}_{\mathbf{x}\in\mathcal{P}} \mathbf{c} \cdot \mathbf{x}$.*

We then show how an edge-space $e$ of $\mathcal{P}$ can be recovered after seeing 3 collinear observed solutions.

**Lemma 3.** *Let $\mathbf{x}, \mathbf{y}, \mathbf{z}$ be 3 distinct collinear points on edges of $\mathcal{P}$. Then they are all on the same edge of $\mathcal{P}$ and the 1-dimensional subspace containing them is an edge-space of $\mathcal{P}$.*

Given the relation between observed solutions and edges, the information $\mathcal{I}^{(t)}$ is stored as follows:

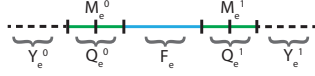

Figure 1: Regions on an edge-space $e$: feasible region $F_e$ (blue), questionable intervals $Q_e^0$ and $Q_e^1$ (green) with their mid-points $M_e^0$ and $M_e^1$ and infeasible regions $Y_e^0$ and $Y_e^1$ (dashed).

**I.1 (Observed Solutions)** `LearnEdge` keeps track of the set of observed solutions that were predicted incorrectly so far $X^{(t)} = \{\mathbf{x}^{(\tau)} : \tau \leq t \quad \hat{\mathbf{x}}^{(\tau)} \neq \mathbf{x}^{(\tau)}\}$ and also the solution for the underlying unknown polytope $\mathbf{x}^* \equiv \operatorname{argmax}_{\mathbf{x}\in\mathcal{P}} \mathbf{c} \cdot \mathbf{x}$ if it is observed.

**I.2 (Edges)** `LearnEdge` keeps track of the set of edge-spaces $E^{(t)}$ given by any 3 collinear points in $X^{(t)}$. For each $e \in E^{(t)}$, it also maintains the regions on $e$ that are certainly *feasible* or *infeasible*. The remaining parts of $e$ called the *questionable* region is where `LearnEdge` cannot classify as infeasible or feasible with certainty (see Figure 1). Formally,

1. **(Feasible Interval)** The *feasible interval* $F_e$ is an interval along $e$ that is identified to be on the boundary of $\mathcal{P}$. More formally, $F_e = \operatorname{Conv}(X^{(t)} \cap e)$.

2. **(Infeasible Region)** The *infeasible region* $Y_e = Y_e^0 \cup Y_e^1$ is the union of two disjoint intervals $Y_e^0$ and $Y_e^1$ that are identified to be outside of $\mathcal{P}$. By Assumption 2, we initialize the infeasible region $Y_e$ to $\{x \in e \mid \|x\|_\infty > 1\}$ for all $e$.

3. **(Questionable Region)** The *questionable region* $Q_e = Q_e^0 \cup Q_e^1$ on $e$ is the union of two disjoint *questionable intervals* along $e$. Formally, $Q_e = e \setminus (F_e \cup Y_e)$. The points in $Q_e$ cannot be certified to be either inside or outside of $\mathcal{P}$ by `LearnEdge`.

4. **(Midpoints in $Q_e$)** For each questionable interval $Q_e^i$, let $M_e^i$ denote the midpoint of $Q_e^i$.

We add the superscript $(t)$ to show the dependence of these quantities on days. Furthermore, we eliminate the subscript $e$ when taking the union over all elements in $E^{(t)}$, e.g. $F^{(t)} = \bigcup_{e \in E^{(t)}} F_e^{(t)}$. So the information $\mathcal{I}^{(t)}$ can be written as follows: $\mathcal{I}^{(t)} = \left(X^{(t)}, E^{(t)}, F^{(t)}, Y^{(t)}, Q^{(t)}, M^{(t)}\right)$.

**Prediction Rules** We now focus on the prediction rules of `LearnEdge`. On day $t$, let $\widetilde{\mathcal{N}}^{(t)} = \{\mathbf{x} \mid \mathbf{p}^{(t)} \cdot \mathbf{x} = b^{(t)}\}$ be the hyperplane specified by the additional constraint $\mathcal{N}^{(t)}$. If $\mathbf{x}^{(t)} \notin \widetilde{\mathcal{N}}^{(t)}$, then $\mathbf{x}^{(t)} = \mathbf{x}^*$ by Corollary 1. So whenever the algorithm observes $\mathbf{x}^*$, it will store $\mathbf{x}^*$ and predict it in the future days when $\mathbf{x}^* \in \mathcal{N}^{(t)}$. This is case **P.1**. So in the remaining cases we know $\mathbf{x}^* \notin \mathcal{N}^{(t)}$.

The analysis of Lemma 2 shows that $\mathbf{x}^{(t)}$ must be in the intersection between $\widetilde{\mathcal{N}}^{(t)}$ and the edges $E_{\mathcal{P}}$, so $\mathbf{x}^{(t)} = \operatorname{argmax}_{\mathbf{x}\in\widetilde{\mathcal{N}}^{(t)}\cap E_{\mathcal{P}}} \mathbf{c} \cdot \mathbf{x}$. Hence, `LearnEdge` can restrict its prediction to the following *candidate* set: $\operatorname{Cand}^{(t)} = \{(E^{(t)} \cup X^{(t)}) \setminus \bar{E}^{(t)}\} \cap \widetilde{\mathcal{N}}^{(t)}$ where $\bar{E}^{(t)} = \{e \in E^{(t)} \mid e \subseteq \widetilde{\mathcal{N}}^{(t)}\}$. As we show in Lemma 4, $\mathbf{x}^{(t)}$ will not be in $\bar{E}^{(t)}$, so it is safe to remove $\bar{E}^{(t)}$ from $\operatorname{Cand}^{(t)}$.

**Lemma 4.** *Let $e$ be an edge-space of $\mathcal{P}$ such that $e \subseteq \widetilde{\mathcal{N}}^{(t)}$, then $\mathbf{x}^{(t)} \notin e$.*

However, $\operatorname{Cand}^{(t)}$ can be empty or only contain points in the infeasible regions of the edge-spaces. If so, then there is simply not enough information to predict a feasible point in $\mathcal{P}$. Hence, `LearnEdge` predicts an arbitrary point outside of $\operatorname{Cand}^{(t)}$. This is case **P.2**.

Otherwise $\mathrm{Cand}^{(t)}$ contains points from the feasible and questionable regions of the edge-spaces. LearnEdge predicts from a subset of $\mathrm{Cand}^{(t)}$ called the *extended feasible region* $\mathrm{Ext}^{(t)}$ instead of directly predicting from $\mathrm{Cand}^{(t)}$. $\mathrm{Ext}^{(t)}$ contains the whole feasible region and only parts of the questionable region on all the edge-spaces in $E^{(t)} \setminus \bar{E}^{(t)}$. We will show later that this guarantees LearnEdge makes progress in learning the true feasible region on some edge-space upon making a mistake. More formally, $\mathrm{Ext}^{(t)}$ is the intersection of $\widetilde{\mathcal{N}}^{(t)}$ with the union of intervals between the two mid-points $(M_e^0)^{(t)}$ and $(M_e^1)^{(t)}$ on every edge-space $e \in E^{(t)} \setminus \bar{E}^{(t)}$ and all points in $X^{(t)}$:
$\mathrm{Ext}^{(t)} = \left\{ X^{(t)} \cup \left\{ \cup_{e \in E^{(t)} \setminus \bar{E}^{(t)}} \mathrm{Conv}\left( (M_e^0)^{(t)}, (M_e^1)^{(t)} \right) \right\} \right\} \cap \widetilde{\mathcal{N}}^{(t)}.$

In **P.3**, if $\mathrm{Ext}^{(t)} \neq \emptyset$ then LearnEdge predicts the point with the highest objective value in $\mathrm{Ext}^{(t)}$.

Finally, if $\mathrm{Ext}^{(t)} = \emptyset$, then we know $\widetilde{\mathcal{N}}^{(t)}$ only intersects within the questionable regions of the learned edge-spaces. In this case, LearnEdge predicts the intersection point with the lowest objective value, which corresponds to **P.4**. Although it might seem counter-intuitive to predict the point with the lowest objective value, this guarantees that LearnEdge makes progress in learning the true feasible region on some edge-space upon making a mistake. The prediction rules are summarized as follows:

**P.1** First, if $\mathbf{x}^*$ is observed and $\mathbf{x}^* \in \mathcal{N}^{(t)}$, then predict $\hat{\mathbf{x}}^{(t)} \leftarrow \mathbf{x}^*$;

**P.2** Else if $\mathrm{Cand} = \emptyset$ or $\mathrm{Cand}^{(t)} \subseteq \bigcup_{e \in E^{(t)}} Y_e^{(t)}$, then predict any point outside $\mathrm{Cand}^{(t)}$;

**P.3** Else if $\mathrm{Ext}^{(t)} \neq \emptyset$, then predict $\hat{\mathbf{x}}^{(t)} = \mathrm{argmax}_{\mathbf{x} \in \mathrm{Ext}^{(t)}} \mathbf{c} \cdot \mathbf{x}$;

**P.4** Else, predict $\hat{\mathbf{x}}^{(t)} = \mathrm{argmin}_{\mathbf{x} \in \mathrm{Cand}^{(t)}} \mathbf{c} \cdot \mathbf{x}$.

**Update Rules**   Next we describe how LearnEdge updates its information. Upon making a mistake, LearnEdge adds $\mathbf{x}^{(t)}$ to the set of previously observed solutions $X^{(t)}$ i.e. $X^{(t+1)} \leftarrow X^{(t)} \cup \{\mathbf{x}^{(t)}\}$. Then it performs one of the following four mutually exclusive update rules (**U.1**-**U.4**) in order.

**U.1** If $\mathbf{x}^{(t)} \notin \widetilde{\mathcal{N}}^{(t)}$, then LearnEdge records $\mathbf{x}^{(t)}$ as the unconstrained optimal solution $\mathbf{x}^*$.

**U.2** Then if $\mathbf{x}^{(t)}$ is not on any learned edge-space in $E^{(t)}$, LearnEdge will try to learn a new edge-space by checking the collinearity of $\mathbf{x}^{(t)}$ and any couple of points in $X^{(t)}$. So after this update LearnEdge might recover a new edge-space of the polytope.

If the previous updates were not invoked, then $\mathbf{x}^{(t)}$ was on some learned edge-space $e$. LearnEdge then compares the objective values of $\hat{\mathbf{x}}^{(t)}$ and $\mathbf{x}^{(t)}$ (we know $\mathbf{c} \cdot \hat{\mathbf{x}}^{(t)} \neq \mathbf{c} \cdot \mathbf{x}^{(t)}$ by Assumption 1):

**U.3** If $\mathbf{c} \cdot \hat{\mathbf{x}}^{(t)} > \mathbf{c} \cdot \mathbf{x}^{(t)}$, then $\hat{\mathbf{x}}^{(t)}$ must be infeasible and LearnEdge then updates the questionable and infeasible regions for $e$.

**U.4** If $\mathbf{c} \cdot \hat{\mathbf{x}}^{(t)} < \mathbf{c} \cdot \mathbf{x}^{(t)}$ then $\mathbf{x}^{(t)}$ was outside of the extended feasible region of $e$. LearnEdge then updates the questionable region and feasible interval on $e$.

In both of **U.3** and **U.4**, LearnEdge will shrink some questionable interval substantially till the interval has length less than $2^{-N}$ in which case Assumption 3 implies that the interval contains no points. So LearnEdge can update the adjacent feasible region and infeasible interval accordingly.

## 3.2   Analysis of LearnEdge

Whenever LearnEdge makes a mistake, one of the update rules **U.1** - **U.4** is invoked. So the number of mistakes of LearnEdge is bounded by the number of times each update rule is invoked. The mistake bound of LearnEdge in Theorem 1 is hence the sum of mistakes bounds in Lemmas 5-7.

**Lemma 5.** *Update U.1 is invoked at most 1 time.*

**Lemma 6.** *Update U.2 is invoked at most $3|E_{\mathcal{P}}|$ times.* [2]

**Lemma 7.** *Updates U.3 and U.4 are invoked at most $O(|E_{\mathcal{P}}|N\log(d))$ times.*

### 3.3 Necessity of the Precision Bound

We show the necessity of Assumption 3 by showing that the dependence on the precision parameter $N$ in our mistake bound is tight. We show that subject to Assumption 3, there exist a polytope and a sequence of additional constraints such that any learning algorithm will make $\Omega(N)$ mistakes. This implies that without any upper bound on precision, it is impossible to learn with finite mistakes.

**Theorem 8.** *For any learning algorithm $\mathcal{A}$ in the Known Objective Problem and any $d \geq 3$, there exists a polytope $\mathcal{P}$ and a sequence of additional constraints $\{\mathcal{N}^{(t)}\}_t$ such that the number of mistakes made by $\mathcal{A}$ is at least $\Omega(N)$.* [3]

### 3.4 Stochastic Setting

Given the lower bound in Theorem 8, we ask "In what settings we can still learn without an upper bound on the precision to which constraints are specified?" The lower bound implies we must abandon the adversarial setting so we consider a PAC style variant. In this variant, the additional constraint at each day $t$ is drawn i.i.d. from some fixed but unknown distribution $\mathcal{D}$ over $\mathbb{R}^d \times \mathbb{R}$ such that each point $(\mathbf{p}, b)$ drawn from $\mathcal{D}$ corresponds to the halfspace $\mathcal{N} = \{\mathbf{x} \mid \mathbf{p} \cdot \mathbf{x} \leq b\}$. We make no assumption on the form of $\mathcal{D}$ and require our bounds to hold in the worst case over all choices of $\mathcal{D}$.

We describe `LearnHull` an algorithm based on the following high level idea: `LearnHull` keeps track of the convex hull $\mathcal{C}^{(t-1)}$ of all the solutions observed up to day $t$. `LearnHull` then behaves as if this convex hull is the entire feasible region. So at day $t$, given the constraint $\mathcal{N}^{(t)} = \{\mathbf{x} \mid \mathbf{p}^{(t)} \cdot \mathbf{x} \leq b^{(t)}\}$, `LearnHull` predicts $\hat{\mathbf{x}}^{(t)}$ where $\hat{\mathbf{x}}^{(t)} = \arg\max_{\mathbf{x} \in \mathcal{C}^{(t-1)} \cap \mathcal{N}^{(t)}} \mathbf{c} \cdot \mathbf{x}$.

`LearnHull`'s hypothetical feasible region is therefore always a subset of the true feasible region – i.e. it can never make a mistake because its prediction was infeasible, but only because its prediction was sub-optimal. Hence, whenever `LearnHull` makes a mistake, it must have observed a point that expands the convex hull. Hence, whenever it fails to predict $\mathbf{x}^{(t)}$, `LearnHull` will enlarge its feasible region by adding the point $\mathbf{x}^{(t)}$ to the convex hull: $\mathcal{C}^{(t)} \leftarrow \text{Conv}(\mathcal{C}^{(t-1)} \cup \{\mathbf{x}^{(t)}\})$, otherwise it will simply set $\mathcal{C}^{(t)} \leftarrow \mathcal{C}^{(t-1)}$ for the next day. We show that the expected number of mistakes of `LearnHull` over $T$ days is linear in the number of edges of $\mathcal{P}$ and only logarithmic in $T$. [4]

**Theorem 9.** *For any $T > 0$ and any constraint distribution $\mathcal{D}$, the expected number of mistakes of `LearnHull` after $T$ days is bounded by $O\left(|E_\mathcal{P}| \log(T)\right)$.*

To prove Theorem 9, first in Lemma 10 we bound the probability that the solution observed at day $t$ falls outside of the convex hull of the previously observed solutions. This is the only event that can cause `LearnHull` to make a mistake. In Lemma 10, we abstract away the fact that the point observed at each day is the solution to some optimization problem.

**Lemma 10.** *Let $\mathcal{P}$ be a polytope and $\mathcal{D}$ a distribution over points on $E_\mathcal{P}$. Let $X = \{x_1, \ldots, x_{t-1}\}$ be $t-1$ i.i.d. draws from $\mathcal{D}$ and $x_t$ an additional independent draw from $\mathcal{D}$. Then $\Pr[x_t \notin \text{Conv}(X)] \leq 2|E_\mathcal{P}|/t$ where the probability is taken over the draws of points $x_1, \ldots, x_t$ from $\mathcal{D}$.*

Finally in Theorem 11 we convert the bound on the expected number of mistakes of `LearnHull` in Theorem 9 to a high probability bound. [5]

**Theorem 11.** *There exists a deterministic procedure such that after $T = O\left(|E_\mathcal{P}| \log\left(1/\delta\right)\right)$ days, the probability (over the randomness of the additional constraint) that the procedure makes a mistake on day $T + 1$ is at most $\delta$ for any $\delta \in (0, 1/2)$.*

## 4 The Known Constraints Problem

We now consider the *Known Constraints Problem* in which the learner observes the changing constraint polytope $\mathcal{P}^{(t)}$ at each day, but does not know the changing objective function which we

assume to be written as $\mathbf{c}^{(t)} = \sum_{i \in S^{(t)}} \mathbf{v}^i$, where $\{\mathbf{v}^i\}_{i \in [n]}$ are fixed but unknown. Given $\mathcal{P}^{(t)}$ and the subset $S^{(t)} \subseteq [n]$, the learner must make a prediction $\hat{\mathbf{x}}^{(t)}$ on each day. Inspired by Bhaskar et al. [5], we use the Ellipsoid algorithm to learn the coefficients $\{\mathbf{v}^i\}_{i \in [n]}$, and show that the mistake bound of the resulting algorithm is bounded by the (polynomial) running time of the Ellipsoid. We use $V \in \mathbb{R}^{d \times n}$ to denote the matrix whose columns are $\mathbf{v}^i$ and make the following assumption on $V$.

**Assumption 6.** *Each entry in $V$ can be written with $N$ bits of precision. Also w.l.o.g. $||V||_F \leq 1$.*

Similar to Section 3 we assume the coordinates of $\mathcal{P}^{(t)}$'s vertices can be written with finite precision.[6]

**Assumption 7.** *The coordinates of each vertex of $\mathcal{P}^{(t)}$ can be written with $N$ bits of precision.*

We first observe that the coefficients of the objective function represent a point that is guaranteed to lie in a region $\mathcal{F}$ (described below) which may be written as the intersection of possibly infinitely many halfspaces. Given a subset $S \subseteq [n]$ and a polytope $\mathcal{P}$, let $\mathbf{x}^{S,\mathcal{P}}$ denote the optimal solution to the instance defined by $S$ and $\mathcal{P}$. Informally, the halfspaces defining $\mathcal{F}$ ensure that for any problem instance defined by arbitrary choices of $S$ and $\mathcal{P}$, the objective value of the *optimal* solution $x^{S,\mathcal{P}}$ must be at least as high as the objective value of any *feasible* point in $\mathcal{P}$. Since the convergence rate of the Ellipsoid algorithm depends on the precision to which constraints are specified, we do not in fact consider a hyperplane for every feasible solution but only for those solutions that are vertices of the feasible polytope $\mathcal{P}$. This is not a relaxation, since LPs always have vertex-optimal solutions.

We denote the set of all vertices of polytope $\mathcal{P}$ by $\mathrm{vert}(\mathcal{P})$, and the set of polytopes $\mathcal{P}$ satisfying Assumption 7 by $\Phi$. We then define $\mathcal{F}$ as follows:

$$\mathcal{F} = \left\{ W = (\mathbf{w}^1, \dots, \mathbf{w}^n) \in \mathbb{R}^{n \times d} \mid \forall S \subseteq [n], \forall \mathcal{P} \in \Phi, \sum_{i \in S} \mathbf{w}^i \cdot \left(\mathbf{x}^{S,\mathcal{P}} - \mathbf{x}\right) \geq 0, \forall \mathbf{x} \in \mathrm{vert}(\mathcal{P}) \right\}$$

The idea behind our `LearnEllipsoid` algorithm is that we will run a copy of the Ellipsoid algorithm with variables $\mathbf{w} \in \mathbb{R}^{d \times n}$, as if we were solving the feasibility LP defined by the constraints defining $\mathcal{F}$. We will always predict according to the centroid of the ellipsoid maintained by the Ellipsoid algorithm (i.e. its candidate solution). Whenever a mistake occurs, we are able to find one of the constraints that define $\mathcal{F}$ such that our prediction violates the constraint – exactly what is needed to take a step in solving the feasibility LP. Since we know $\mathcal{F}$ is non-empty (at least the true objective function $V$ lies within it) we know that the LP we are solving is feasible. Given the polynomial convergence time of the Ellipsoid algorithm, this gives a polynomial mistake bound for our algorithm.

The Ellipsoid algorithm will generate a sequence of ellipsoids with decreasing volume such that each one contains feasible region $\mathcal{F}$. Given the ellipsoid $\mathcal{E}^{(t)}$ at day $t$, `LearnEllipsoid` uses the centroid of $\mathcal{E}^{(t)}$ as its hypothesis for the objective function $W^{(t)} = \left((\mathbf{w}^1)^{(t)}, \dots, (\mathbf{w}^n)^{(t)}\right)$. Given the subset $S^{(t)}$ and polytope $\mathcal{P}^{(t)}$, `LearnEllipsoid` predicts $\hat{\mathbf{x}}^{(t)} \in \mathrm{argmax}_{\mathbf{x} \in \mathcal{P}^{(t)}} \{\sum_{i \in S^{(t)}} (\mathbf{w}^i)^{(t)} \cdot \mathbf{x}\}$. When a mistake occurs, `LearnEllipsoid` finds the hyperplane $\mathcal{H}^{(t)} = \left\{ W = (\mathbf{w}^1, \dots, \mathbf{w}^n) \in \mathbb{R}^{n \times d} : \sum_{i \in S^{(t)}} \mathbf{w}^i \cdot (\mathbf{x}^{(t)} - \hat{\mathbf{x}}^{(t)}) > 0 \right\}$ that separates the centroid of the current ellipsoid (the current candidate objective) from $\mathcal{F}$.

After the update, we use the Ellipsoid algorithm to compute the minimum-volume ellipsoid $\mathcal{E}^{(t+1)}$ that contains $\mathcal{H}^{(t)} \cap \mathcal{E}^{(t)}$. On day $t + 1$, `LearnEllipsoid` sets $W^{(t+1)}$ to be the centroid of $\mathcal{E}^{(t+1)}$.

We left the procedure used to solve the LP in the prediction rule of `LearnEllipsoid` unspecified. To simplify our analysis, we use a specific LP solver to obtain a prediction $\hat{\mathbf{x}}^{(t)}$ which is a vertex of $\mathcal{P}^{(t)}$.

**Theorem 12** (Theorem 6.4.12 and Remark 6.5.2 [11])**.** *There exists a LP solver that runs in time polynomial in the length of its input and returns an exact solution that is a vertex of $\mathcal{P}^{(t)}$.*

In Theorem 13, we show that the number of mistakes made by `LearnEllipsoid` is at most the number of updates that the Ellipsoid algorithm makes before it finds a point in $\mathcal{F}$ and the number of updates of the Ellipsoid algorithm can be bounded by well-known results from the literature on LP.

**Theorem 13.** *The total number of mistakes and the running time of `LearnEllipsoid` in the Known Constraints Problem is at most $\mathrm{poly}(n, d, N)$.*

## Footnotes

[1] Lemma 6.2.4 from Grotschel et al. [11] states that if each constraint in $\mathcal{P} \subseteq \mathbb{R}^d$ has encoding length at most $N$ then each vertex of $\mathcal{P}$ has encoding length at most $4d^2N$. Typically the finite precision assumption is made on the constraints of the LP. However, since this assumption implies that the vertices can be described with finite precision, for simplicity, we make our assumption directly on the vertices.

[2]The dependency on $|E_{\mathcal{P}}|$ can be improved by replacing it with the set of edges of $\mathcal{P}$ on which an optimal solution is observed. This applies to all the dependencies on $|E_{\mathcal{P}}|$ in our bounds.

[3] We point out that the condition $d \geq 3$ is necessary in the statement of Theorem 8 since there exists learning algorithms for $d = 1$ and $d = 2$ with finite mistake bounds independent of $N$. See the supplementary material.

[4] `LearnHull` can be implemented efficiently in time $\text{poly}(T, N, d)$ if all of the coefficients in the unknown constraints in $\mathcal{P}$ are represented in $N$ bits. Note that given the observed solutions so far and a new point, a separation oracle can be implemented in time $\text{poly}(T, N, d)$ using a LP solver.

[5] `LearnEdge` fails to give any non-trivial mistake bound in the adversarial setting.

[6]We again point out that this is implied if the halfspaces defining the polytope are described with finite precision [11].

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
