[Supplementary Material · 866-learning-from-rational-behavior-supplementary.pdf]

# Supplementary material for "Learning from Rational Behavior: Predicting Solutions to Unknown Linear Programs"

**Shahin Jabbari, Ryan Rogers, Aaron Roth, Zhiwei Steven Wu**

## A Polynomial Mistake Bound with Exponential Running Time

In this section we give a simple randomized algorithm for the unknown constraints problem, that in expectation makes a number of mistakes that is only linear in the dimension $d$, the number of rows in the unknown constraint matrix $A$ (denoted by $m$), and the bit precision $N$, but which requires exponential running time. When the number of rows is large, this can represent an exponential improvement over the mistake bound of LearnEdge, which is linear in the number of *edges* on the polytope $\mathcal{P}$ defined by $A$. This algorithm which we describe shortly is a randomized variant of the well known halving algorithm [13]. We leave it as an open problem whether the mistake bound achieved by this algorithm can also be achieved by a computationally efficient algorithm.

Let $\mathcal{K}$ be the hypothesis class of all polytopes formed by $m$ constraints in $d$ dimensions, such that each entry of each constraint can be written as a multiple of $1/2^N$ (and without loss of generality, up to scaling, has absolute value at most 1). We then have

$$|\mathcal{K}| = 2^{O(dmN)}.$$

We write $\mathcal{K}^{(t)}$ to denote the polytopes that are consistent with the examples and solutions we have seen up to and including day $t$. Note that $|\mathcal{K}^{(t)}| \geq 1$ for every $t$ because there is some polytope (specifically the true unknown polytope $\mathcal{P}$) that is consistent with all the optimal solutions. On each day $t$ we keep track of consistent polytopes and more specifically update the set of consistent polytopes by

$$\mathcal{K}^{(t+1)} = \left\{ \mathcal{P} \in \mathcal{K}^{(t)} \mid \mathbf{x}^{(t)} \in \underset{\mathbf{x} \in \mathcal{P} \cap \mathcal{N}^{(t)}}{\operatorname{argmax}} \mathbf{c} \cdot \mathbf{x} \right\}, \tag{3}$$

where $\mathcal{N}^{(t)}$ is the new constraint on day $t$. The formal description of the algorithm, FCP, is presented in Algorithm 1. To predict at each day, FCP selects a polytope $\hat{\mathcal{P}}^{(t)}$ from $\mathcal{K}^{(t)}$ uniformly at random and guesses $\hat{\mathbf{x}}^{(t)}$ that solves the following LP: $\max_{\mathbf{x} \in \hat{\mathcal{P}}^{(t)} \cap \mathcal{N}^{(t)}} \mathbf{c} \cdot \mathbf{x}$.

---
**Algorithm 1** Find Consistent Polytope FCP

> **procedure** FCP
>> $\mathcal{K}^{(1)} = \mathcal{K}$.        ▷ Initialize
>> **for** $t = 1 \ldots$ **do**
>>> Choose $\hat{\mathcal{P}}^{(t)} \in \mathcal{K}^{(t)}$ uniformly at random.
>>> Guess $\hat{\mathbf{x}}^{(t)} \in \operatorname{argmax}_{\mathbf{x} \in \hat{\mathcal{P}}^{(t)} \cap \mathcal{N}^{(t)}} \mathbf{c} \cdot \mathbf{x}$.        ▷ Predict
>>> Observe $\mathbf{x}^{(t)}$ and set $\mathcal{K}^{(t+1)}$ as in (3).
> **end procedure**

---

We now bound the expected number of mistakes that FCP makes.

**Theorem 14.** *The expected number of mistakes that FCP makes is at most* $\log(|\mathcal{K}|) = O(dmN)$, *where the expectation is over the randomness of FCP and possible randomness of the adversary.*

*Proof.* First note that the probability that FCP *does not* make a mistake at day $t$ can be expressed as $|\mathcal{K}^{(t+1)}|/|\mathcal{K}^{(t)}|$. This is because if FCP makes a mistake at day $t$, it must have selected a polytope that will be eliminated at the next day (also note that FCP selects its polytope from among the consistent set uniformly at random). Now consider the product of these probabilities over all days $t = 1 \ldots T$.

$$\prod_{t=1}^{T} (1 - \mathbb{P}\left[\text{Mistake at day } t\right]) = \prod_{t=1}^{T} \frac{|\mathcal{K}^{(t+1)}|}{|\mathcal{K}^{(t)}|} = \frac{|\mathcal{K}^{(T+1)}|}{|\mathcal{K}^{(1)}|}.$$

Finally, note that the expected number of mistakes is the sum of probabilities of making mistakes over all days. Using the inequality $(1 - x) \leq e^{-x}$ for every $x \in [0, 1]$ and rearranging terms we get

$$\sum_{t=1}^{T} \mathbb{P}\left[\text{Mistake at day } t\right] \leq \log\left(\frac{|\mathcal{K}^{(1)}|}{|\mathcal{K}^{(T+1)}|}\right) \leq O(dmN),$$

since $|\mathcal{K}^{(1)}| = 2^{O(dmN)}$ and $|\mathcal{K}^{(T+1)}| \geq 1$. $\qquad\square$

Finally, we remark that the randomized halving technique above will also result in a polynomial mistake bound in the more demanding variant where not only the underlying constraint matrix but also the linear objective function is unknown. This is because the coefficients of the objective function can be written in $dN$ bits if they are also represented with finite precision. However, the issue about the exponential running time still exists in the new setting.

## B   Missing Proofs from Section 3

### B.1   Section 3.1

**Proof of Lemma 2.** Let $\mathbf{x}^*$ be the optimal solution of the linear program solved over the unknown polytope $\mathcal{P}$, without the added constraint i.e. $\mathbf{x}^* \equiv \operatorname{argmax}_{x \in \mathcal{P}} \mathbf{c} \cdot \mathbf{x}$.

1. Suppose that $\mathbf{x}^* \in \mathcal{N}^{(t)}$, then clearly $\mathbf{x}^{(t)} = \mathbf{x}^*$. By Assumption 1, $\mathbf{x}^*$ lies on a vertex of $\mathcal{P}$ and therefore $\mathbf{x}^{(t)}$ lies on one of the edges of $\mathcal{P}$.

2. Suppose that $\mathbf{x}^* \notin \mathcal{N}^{(t)}$ i.e. $\mathbf{p}^{(t)} \cdot \mathbf{x}^* > b^{(t)}$. Then we claim that the optimal solution $\mathbf{x}^{(t)}$ satisfies $\mathbf{p}^{(t)} \cdot \mathbf{x}^{(t)} = b^{(t)}$. Suppose to the contrary that $\mathbf{p}^{(t)} \cdot \mathbf{x}^{(t)} < b^{(t)}$. Since $\mathbf{c} \cdot \mathbf{x}^* \geq \mathbf{c} \cdot \mathbf{x}^{(t)}$, then for any point $\mathbf{y} \in \operatorname{Conv}(\mathbf{x}^{(t)}, \mathbf{x}^*)$,

$$\mathbf{c} \cdot \mathbf{y} = \mathbf{c} \cdot (\alpha \mathbf{x}^{(t)} + (1 - \alpha)\mathbf{x}^*) = \alpha(\mathbf{c} \cdot \mathbf{x}^{(t)}) + (1 - \alpha)(\mathbf{c} \cdot \mathbf{x}^*) \geq \mathbf{c} \cdot \mathbf{x}^{(t)} \qquad \forall \alpha \in [0, 1].$$

Since $\mathbf{x}^{(t)}$ strictly satisfies the new constraint, there exists some point $\mathbf{y}^* \in \operatorname{Conv}(\mathbf{x}^{(t)}, \mathbf{x}^*)$ where $\mathbf{y}^* \neq \mathbf{x}^{(t)}$ such that $\mathbf{y}^* \in \mathcal{P}^{(t)}$ (i.e. $\mathbf{y}^*$ is also feasible). It follows that $\mathbf{c} \cdot \mathbf{y}^* \geq \mathbf{c} \cdot \mathbf{x}^{(t)}$, which contradicts Assumption 1. Therefore, $\mathbf{x}^{(t)}$ must bind the additional constraint. Furthermore, by non-degeneracy Assumption 5, $\mathbf{x}^{(t)}$ binds exactly $(d - 1)$ constraints in $\mathcal{P}$, i.e. $\mathbf{x}^{(t)}$ lies at the intersection of $d - 1$ hyperplanes of $\mathcal{P}$ which are linearly independent by Assumption 4. Therefore, $\mathbf{x}^{(t)}$ must be on an edge of $\mathcal{P}$.

$\qquad\square$

**Proof of Lemma 3.** Without loss of generality, let us assume $\mathbf{y}$ can be written as convex combination of $\mathbf{x}$ and $\mathbf{z}$ i.e. $\mathbf{y} = \alpha \mathbf{x} + (1 - \alpha)\mathbf{z}$ for some $\alpha \in (0, 1)$. Let $B_y = \{j \mid A_j \mathbf{y} = \mathbf{b}_j\}$ be the set of binding constraints for $\mathbf{y}$. We know that $|B_y| \geq d - 1$ by Assumption 5. For any $j$ in $B_y$, we consider the following two cases.

1. At least one of $\mathbf{x}$ and $\mathbf{z}$ belongs to the hyperplane $\{\mathbf{w} \mid A_j \mathbf{w} = \mathbf{b}_j\}$. Then we claim that all three points bind the same constraint. Assume that $A_j \mathbf{x} = \mathbf{b}_j$, then we must have

$$A_j \mathbf{z} = \frac{A_j(\mathbf{y} - \alpha \mathbf{x})}{(1 - \alpha)} = \frac{\mathbf{b}_j - \alpha \mathbf{b}_j}{(1 - \alpha)} = \mathbf{b}_j.$$

Similarly, if we assume $A_j \mathbf{z} = \mathbf{b}_j$, we will also have $A_j \mathbf{x} = \mathbf{b}_j$.

2. None of $\mathbf{x}$ and $\mathbf{z}$ belongs to the hyperplane $\{\mathbf{w} \mid A_j \mathbf{w} = \mathbf{b}_j\}$ i.e. $A_j \mathbf{x} < \mathbf{b}_j$ and $A_j \mathbf{z} < \mathbf{b}_j$ both hold. Then we can write

$$\mathbf{b}_j = A_j \mathbf{y} = \alpha A_j \mathbf{x} + (1 - \alpha)A_j \mathbf{z} < \alpha \mathbf{b}_j + (1 - \alpha)\mathbf{b}_j = \mathbf{b}_j,$$

which is a contradiction.

It follows that for any $j \in B_y$, we have $A_j \mathbf{x} = A_j \mathbf{y} = A_j \mathbf{z} = \mathbf{b}_j$. Since $|B_y| \geq d - 1$, we know by Assumption 4 that the set of points that bind any set of $d - 1$ constraints in $B_y$ will form an edge-space and further this edge-space will include $\mathbf{x}, \mathbf{y}$, and $\mathbf{z}$. $\qquad\square$

**Proof of Lemma 4.** First, note that the observed solution $\mathbf{x}^{(t)}$ is a vertex in the polytope $\mathcal{P}^{(t)} = \mathcal{P} \cap \widetilde{\mathcal{N}}^{(t)}$, that is an intersection of *exactly* $d$ constraints by Assumption 1 and Assumption 5. Second, note that all points in $e$ bind at least $d-1$ constraints in $\mathcal{P}$ and since $e \subseteq \widetilde{\mathcal{N}}^{(t)}$, then all points in $e$ bind at least $d$ constraints in $\mathcal{P}^{(t)}$. It follows that any vertex of $\mathcal{P}^{(t)}$ on $e$ must bind at least $(d+1)$ constraints, which rules out the possibility of $\mathbf{x}^{(t)}$ being on $e$. $\qquad\square$

## B.2 Section 3.2

**Proof of Lemma 5.** As soon as `LearnEdge` invokes update rule **U.1**, it records the solution $\mathbf{x}^* \equiv \text{argmax}_{\mathbf{x} \in \mathcal{P}} \, \mathbf{c} \cdot \mathbf{x}$. Then, the prediction rule specified by **P.1** prevents further updates of this type. This is because $\mathbf{x}^*$ continues to remain optimal if it feasible in the more constrained problem (optimizing over the polytope $\mathcal{P}^{(t)}$). $\qquad\square$

**Proof of Lemma 6.** Rule **U.2** is invoked only when $\mathbf{x}^{(t)} \notin X^{(t)}$ and $\mathbf{x}^{(t)} \notin e$ for any of $e \in E^{(t)}$. So after each invokation, a new point on the edge of $\mathcal{P}$ is observed. Whenever 3 points are observed on the same edge of $\mathcal{P}$, the edge-space is learned by Lemma 3 (since the points are necessarily collinear). Hence, the total number of times rule **U.2** can be invoked is at most $3|E_{\mathcal{P}}|$. $\qquad\square$

We now introduce Lemmas 16 and 17 that will be used in the proof of Lemma 18 which itself will be useful in the proof of Lemma 7. But first, for completeness, in Lemma 15 we show that we are guaranteed the existence of an edge-space if the update implemented is **U.3** or **U.4**.

**Lemma 15.**

    *(1) If update rule **U.3** is used, then there exists edge-space $\hat{e} \in E^{(t)}$ such that $\hat{\mathbf{x}}^{(t)} \in \hat{e}$.*

    *(2) If update rule **U.4** is used, then there exists edge-space $e \in E^{(t)}$ such that $\mathbf{x}^{(t)} \in e$.*

*Proof.* We prove this by contradiction. First consider the case in which $\mathbf{c} \cdot \hat{\mathbf{x}}^{(t)} > \mathbf{c} \cdot \mathbf{x}^{(t)}$ and suppose $\hat{\mathbf{x}}^{(t)} \in \{\mathbf{x} \in X^{(t)} \mid \forall e \in E^{(t)}, \mathbf{x} \notin e\}$. When this is the case we know that $\hat{\mathbf{x}}^{(t)}$ is feasible at day $t$ and this contradicts $\mathbf{x}^{(t)}$ being optimal at that day because $\mathbf{c} \cdot \hat{\mathbf{x}}^{(t)} > \mathbf{c} \cdot \mathbf{x}^{(t)}$.

Next consider the case in which $\mathbf{c} \cdot \hat{\mathbf{x}}^{(t)} < \mathbf{c} \cdot \mathbf{x}^{(t)}$ and suppose $\mathbf{x}^{(t)} \in \{\mathbf{x} \in X^{(t)} \mid \forall e \in E^{(t)}, \mathbf{x} \notin e\}$. We would have used **P.3** to make a prediction because $\widetilde{\mathcal{N}}^{(t)} \cap \text{Ext}^{(t)}$ is non-empty and includes at least the point $\mathbf{x}^{(t)}$. Note that by **P.3**, we have $\hat{\mathbf{x}}^{(t)} = \text{argmax}_{\widetilde{\mathcal{N}}^{(t)} \cap \text{Ext}^{(t)}} \, \mathbf{c} \cdot \mathbf{x}$. Since $\mathbf{x}^{(t)} \in \widetilde{\mathcal{N}}^{(t)} \cap \text{Ext}^{(t)}$, we must also have $\mathbf{c} \cdot \hat{\mathbf{x}}^{(t)} \geq \mathbf{c} \cdot \mathbf{x}^{(t)}$, which is again a contradiction. $\qquad\square$

**Lemma 16.** *If **U.3** is implemented at day $t$, then $\hat{\mathbf{x}}^{(t)} \notin \mathcal{P}$ and $\hat{\mathbf{x}}^{(t)} \in (Q_{\hat{e}}^i)^{(t)} \cap \text{Ext}^{(t)}$ for some $i = 0$ or $1$ where $\hat{e}$ is given in **U.3**.*

*Proof.* Each time the algorithm makes update **U.3** we know that the algorithm's prediction $\hat{\mathbf{x}}^{(t)}$ was on some edge-space $\hat{e} \in E^{(t)}$ by Lemma 15. Therefore, `LearnEdge` did not use **P.1** or **P.2** to predict $\hat{\mathbf{x}}^{(t)}$. So we only need to check **P.3** and **P.4**.

- If **P.3** was used, we know that $\hat{\mathbf{x}}^{(t)} \in \widetilde{\mathcal{N}}^{(t)}$ but $\hat{\mathbf{x}}^{(t)}$ must violate a constraint of $\mathcal{P}$, due to $\mathbf{x}^{(t)}$ being the observed solution and having lower objective value. This implies that $\hat{\mathbf{x}}^{(t)}$ is in some questionable region, say $(Q_{\hat{e}}^i)^{(t)}$ for $i = 0$ or $1$ but also in the extended feasible on $\hat{e}$, i.e. $\hat{\mathbf{x}}^{(t)} \in \text{Ext}^{(t)} \cap (Q_{\hat{e}}^i)^{(t)}$.

- If **P.4** was used, then $\text{Ext}^{(t)} = \emptyset$. However `LearnEdge` selected $\hat{\mathbf{x}}^{(t)}$ from $\text{Cand}^{(t)} \neq \emptyset$ with the lowest objective value. Finally, when updating with **U.3** *(i)* $\mathbf{x}^{(t)} \in \text{Cand}^{(t)}$ and *(ii)* $\mathbf{c} \cdot \mathbf{x}^{(t)} < \mathbf{c} \cdot \hat{\mathbf{x}}^{(t)}$. So we could not have used **P.4** to predict $\hat{\mathbf{x}}^{(t)}$.

$\qquad\square$

**Lemma 17.** *If **U.4** is implemented at day $t$, then $\mathbf{x}^{(t)} \in (Q_e^i)^{(t)} \backslash \text{Ext}^{(t)}$ for some $i = 0$ or $1$ where $e$ is given in **U.4**.*

*Proof.* As in Lemma 16, `LearnEdge` did not use **P.1** or **P.2** to predict $\hat{\mathbf{x}}^{(t)}$ (again by application of Lemma 15). So we only need to check **P.3** and **P.4**.

- If **P.3** was used, then `LearnEdge` did not guess $\mathbf{x}^{(t)}$ which had the higher objective because it was outside of $\text{Ext}^{(t)}$ along edge-space $e$. Since $\mathbf{x}^{(t)}$ is feasible, it must have been on some questionable region on $e$, say $(Q_e^i)^{(t)}$ for some $i = 0$ or $1$. Hence, $\mathbf{x}^{(t)} \in (Q_e^i)^{(t)} \backslash \text{Ext}^{(t)}$.

- If **P.4** was used, then $\text{Ext}^{(t)} = \emptyset$ and thus $\mathbf{x}^{(t)}$ was a candidate solution but outside of the extended feasible interval along edge-space $e$. Further, because $\mathbf{x}^{(t)} \in \mathcal{P}$ we know that $\mathbf{x}^{(t)}$ must be in some questionable interval along $e$, say $(Q_e^i)^{(t)}$ for some $i = 0$ or $1$. Therefore, $\mathbf{x}^{(t)} \in (Q_e^i)^{(t)} \backslash \text{Ext}^{(t)}$.

$\square$

**Lemma 18.** *Each time **U.3** or **U.4** is used, there is a questionable interval on some edge-space whose length is decreased by at least a factor of two.*

*Proof.* From Lemma 16 we know that if **U.3** is used then $\hat{\mathbf{x}}^{(t)} \in \hat{e}$, is infeasible but outside of the known infeasible interval $(Y_{\hat{e}}^i)^{(t)}$ and inside of the extended feasible interval along $\hat{e}$. Note that if a point $\mathbf{x}$ is infeasible along edge space $\hat{e}$ in the questionable interval $(Q_{\hat{e}}^i)^{(t)}$, then the constraint it violates is also violated by all points in $Y_e^i$. Hence the interval $\text{Conv}(\mathbf{x}, (Y_e^i)^{(t)})$ contains only infeasible points. By the definition of $(M_e^i)^{(t)}$ and the fact that $\hat{\mathbf{x}}^{(t)}$ is in the extended feasible region on $\hat{e}$, we know that

$$|(Q_e^i)^{(t+1)}| = \left|(Q_e^i)^{(t)} \backslash \text{Conv}\left(\hat{\mathbf{x}}^{(t)}, (Y_e^i)^{(t)}\right)\right| \leq \left|(Q_e^i)^{(t)} \backslash \text{Conv}\left((M_e^i)^{(t)}, (Y_e^i)^{(t)}\right)\right| = \frac{|(Q_e^i)^{(t)}|}{2}.$$

Further, from convexity we know that if $\mathbf{x}^{(t)}$ is feasible on edge-space $e$ at day $t$, then the interval $\text{Conv}(\mathbf{x}^{(t)}, F_e^{(t)})$ only contains feasible points on $e$. We know that $\mathbf{x}^{(t)}$ is feasible and in a questionable interval $(Q_e^i)^{(t)}$ along edge space $e$ but outside its extended feasible region, by Lemma 17. Thus, by definition of the midpoint $(M_e^i)^{(t)}$ we have

$$|(Q_e^i)^{(t+1)}| = \left|(Q_e^i)^{(t)} \backslash \text{Conv}\left(\hat{\mathbf{x}}^{(t)}, (F_e)^{(t)}\right)\right| \leq \left|(Q_e^i)^{(t)} \backslash \text{Conv}\left((M_e^i)^{(t)}, F_e^{(t)}\right)\right| = \frac{|(Q_e^i)^{(t)}|}{2}.$$

$\square$

**Proof of Lemma 7.** Let $Q_e^i$ be the updated questionable interval. We know initially $Q_e^i$ has length at most than $2\sqrt{d}$ by Assumption 2. In Lemma 18 we showed that each time an update **U.3** or **U.4** is invoked, the length of $Q_e^i$ is decreases by at least a half. Then after at most $O(N \log(d))$ updates, the interval will have length less than $2^{-N}$ after which the interval will be updated at most once because there is at most one point up to precision $N$ in it.

Therefore, the total number of updates on $Q_e^i$ is bounded by $O(N \log(d))$. Since there are at most $2|E_{\mathcal{P}}|$ questionable intervals, the total number of updates **U.3** and **U.4** is bounded by $O(|E_{\mathcal{P}}| N \log(d))$. $\square$

### B.3 Section 3.3

We prove the lower bound in Theorem 8 initially for $d = 3$.

**Theorem 19.** *If Assumptions 1 and 3 hold, then the number of mistakes of any learning algorithm in the known objective problem is at least $\Omega(N)$ for $d = 3$.*

*Proof.* The high level idea of the proof is as follows. In each day the adversary can pick two points on the two bold edges in Figure 2 as the optimal points and no matter what the learner predicts, the adversary can return a point that is different than the guess of the learner as the optimal point. If the adversary picks the midpoint of the questionable region in each day, then the size of the questionable region in both of the lines will shrink in half. So this process can be repeated $N$ times where each entry of every vertex can be written with as a multiple of $1/2^N$, by Assumption 3. Finally, we show

that at the end of this process, the adversary can return a simple polytope which is consistent with all the observed optimal points so far.

We formalize this high level in procedure ADVERSARY that takes as input any learning algorithm $\mathcal{L}$ and interacts with $\mathcal{L}$ for $N$ days. Each day the adversary presents a constraint. Then no matter what $\mathcal{L}$ predicts, the adversary ensures that $\mathcal{L}$'s prediction is incorrect. After $N$ interactions, the adversary outputs a feasible polytope that is consistent with all of the previous actions of the adversary.

Figure 2: The underlying polytope in the proof of Theorem 8. The two learned edges are in bold.

In procedure ADVERSARY, subroutines NAC and AD-2 are used to pick a constraint and return an optimal point that causes $\mathcal{L}$ to make a mistake, respectively. We use the notation $\mathrm{mid}(R)$ in subroutines NAC and AD-2 to denote the middle point of a real interval $R$, $\mathrm{top}(R)$ to be the largest point in $R$, and $\mathrm{bot}(R)$ to be the smallest value in $R$. Finally, we assume the known objective function is $c = (0, 0, 1)$.

---

**Algorithm 2** Adversary Updates (ADVERSARY)
___
**Input:** Any learning algorithm $\mathcal{L}$ and bit precision $N$
**Output:** Polytope $\mathcal{P}$ that is consistent with $\mathcal{L}$ making a mistake each day.
  **procedure** ADVERSARY($\mathcal{L}, N$)
      Set $R_1^{(0)} = [0, 1], R_2^{(0)} = [1, 2]$.                                           ▷ Initialize
      **for** $t = 1, \cdots, N$ **do**
          $\left( \left( \mathbf{p}^{(t)}, q^{(t)} \right), r_1^{(t)}, r_2^{(t)} \right) \leftarrow \text{NAC}(R_1^{(t-1)}, R_2^{(t-1)})$.
          Show constraint $\mathbf{p}^{(t)} \cdot \mathbf{x} \le q^{(t)}$ to $\mathcal{L}$.            ▷ Constraint
          Get prediction $\hat{\mathbf{x}}^{(t)}$ from $\mathcal{L}$.
          $\left( \mathbf{x}^{(t)}, R_1^{(t)}, R_2^{(t)} \right) \leftarrow \text{AD-2} \left( R_1^{(t-1)}, R_2^{(t-1)}, r_1^{(t)}, r_2^{(t)}, \hat{\mathbf{x}}^{(t)} \right)$.     ▷ Update
          Reveal the optimal $\mathbf{x}^{(t)} \ne \hat{\mathbf{x}}^{(t)}$ and update the regions $R_1^{(t)}$ and $R_2^{(t)}$.
    $A, \mathbf{b} \leftarrow \text{MATRIX}(R_1^N, R_2^N)$        ▷ Constraint matrix consistent with $\{\mathbf{x}^{(t)} \mid t \in [N]\}$
  **return** $A, \mathbf{b}$
  **end procedure**

---

The procedure NAC takes as input two real valued intervals and then outputs two points $r_1$ and $r_2$ as well as the new constraint denoted by the pair $(\mathbf{p}, q)$. The two points will be used as input in AD-2 along with the learner's prediction. In procedure AD-2 the adversary makes sure that the learner suffers a mistake. On each day, one of the points say $r_2$ produced by NAC has a higher objective than the other one. If the learner chooses $r_2$ then the adversary will simply choose a polytope that makes $r_2$ infeasible so that $r_1$ is actually the optimal point that day. If the learner chooses $r_1$ then the adversary picks $r_2$ as the optimal solution. Note that the three points $r_1$, $r_2$, and $r_3$ computed in NAC all bind the constraint and are not collinear, and thus uniquely define the hyperplane $\{\mathbf{x} : \mathbf{p} \cdot \mathbf{x} = q\}$. Finally, in AD-2 the adversary updates the new feasible region for her use in the next days.

---

**Algorithm 3** New Adversarial Constraint (`NAC`)

---

**procedure** NAC($R_1, R_2$)
    Set $\epsilon \leftarrow 0.01$.
    Set $r_1 \leftarrow (0, 1, \text{mid}(R_1))$
        $r_2 \leftarrow (1, \text{mid}(R_2), 1 + \epsilon \cdot \text{mid}(R_2))$
        $r_3 \leftarrow (1, \text{mid}(R_2), 0)$.
    Set $\mathbf{p} = (1 - \text{mid}(R_2), 1, 0)$ and $q = 1$    ▷ The constraint is $\mathbf{p} \cdot \mathbf{x} \leq 1$ and binds at $r_1, r_2, r_3$
    **return** $(\mathbf{p}, q)$ and $r_1, r_2$.
**end procedure**

---

---

**Algorithm 4** Adaptive Adversary (`AD-2`)

---

**procedure** AD-2($R_1, R_2, r_1, r_2, \hat{\mathbf{x}}$)
    **if** $\hat{\mathbf{x}} == r_2$ **then**                                ▷ $r_1$ and $r_2$ as in Algorithm 3
        $\mathbf{x} \leftarrow r_1$.
        $R_2 \leftarrow [\text{bot}(R_2), \text{mid}(R_2)]$.
    **else**
        $\mathbf{x} = r_2$.
        $R_2 \leftarrow [\text{mid}(R_2), \text{top}(R_2)]$.
    $R_1 \leftarrow [\text{mid}(R_1), \text{top}(R_1)]$.
    **return** $\mathbf{x}, R_1, R_2$
**end procedure**

---

`ADVERSARY` finishes by actually outputting the polytope that was consistent with the constraints and the optimal solutions he showed at each day. This polytope is defined by constraint matrix $A$ and vector $\mathbf{b}$ using the subroutine `MATRIX` as well as the nonnegativity constraint $\mathbf{x} \geq 0$.

---

**Algorithm 5** Matrix consistent with adversary (`MATRIX`)

---

**procedure** MATRIX($R_1, R_2$)
    Set $f_1 = (\text{top}(R_1) + \text{bot}(R_1))/2$ and $f_2 = (\text{top}(R_2) + \text{bot}(R_2))/2$ and $\epsilon > 0$

$$A \leftarrow \begin{pmatrix} -1 & 0 & 0 \\ 1 & 0 & 0 \\ (f_1 - 1 - \epsilon) & -\epsilon & 1 \\ -(f_2 - 1) \cdot f_1 & f_1 & 0 \end{pmatrix} \text{ and } \mathbf{b} \leftarrow \begin{pmatrix} 0 \\ 1 \\ f_1 - \epsilon \\ f_1 \end{pmatrix}.$$

    **return** $A$ and $\mathbf{b}$
**end procedure**

---

To prove that the procedure given in `ADVERSARY` does in fact make every learner $\mathcal{L}$ make a mistake at every day, we need to show that *(i)* there exists a simple unknown polytope that is consistent with what the adversary has presented in the previous days. Furthermore, we need to show that *(ii)* the optimal point returned by the adversary on each day is indeed the optimal point corresponding to the LP with objective $\mathbf{c}$ and unknown constraints subject to the additional constraint added on each day.

To show *(i)* note that point $r_1^{(t)} = (0, 1, \text{mid}(R_1^{(t-1)}))$ is always a feasible point for $t \in [N]$ in the polytope given by $A$ and $\mathbf{b}$ and the new constraint added each day will allow $r_1^{(t)}$ to remain feasible.

To show *(ii)* first note that the new constraint added is always a binding constraint. So by Assumption 5, it is sufficient to check the intersection of the edges of the polytope output by `MATRIX` and the newly added hyperplane and return the (feasible) point with the highest objective as the optimal point. Second, the following equations define the edges of the polytope which are one dimensional subspaces $\mathbf{e}^{i,j}$ according to Assumption 4 with $A$ and $\mathbf{b}$ being the output of `MATRIX`.

$$\mathbf{e}^{i,j} = \{\mathbf{x} \in \mathbb{R}^3 \mid A_i \mathbf{x} = b_i \quad \text{and} \quad A_j \mathbf{x} = b_j\} \qquad i, j \in \{1, 2, 3, 4\}, \quad i \neq j,$$

where $A_i$ is the $i$th row of $A$. Since the first two constraints define two parallel hyperplanes, we only need to consider 5 edges. Let

$$r_1^{(t)} = \left(0, 1, \text{mid}(R_1^{(t-1)})\right),$$

and

$$r_2^{(t)} = \left(1, \mathrm{mid}(R_2^{(t-1)}), 1 + \epsilon \cdot \mathrm{mid}(R_2^{(t-1)})\right).$$

We show that the new constraint either intersects the edges of the polytope at $r_1^{(t)}$ or $r_2^{(t)}$ or do not intersect with them at all. This will prove that the optimal points shown by the adversary each day is consistent with the unknown polytope.

1. $\mathbf{e}^{1,4} = (0, 0, f_1) \cdot s + (0, 1, 0)$ that intersects with the new hyperplane at $r_1^{(t)}$.

2. $\mathbf{e}^{2,3} = (0, f_2, \epsilon \cdot f_2) \cdot s + (1, 0, 1)$ that intersects with the new hyperplane at $r_2^{(t)}$.

3. $\mathbf{e}^{2,4} = (0, 0, 1 + \epsilon \cdot f_2) \cdot s + (1, f_2, 0)$ that does not intersect the new hyperplane unless $\mathrm{mid}(R_2) = f_2$ (which does not happen).

4. $\mathbf{e}^{3,4} = (1, f_2 - 1, 1 + \epsilon \cdot f_2 - f_1) \cdot s + (0, 1, f_1)$ that does not intersect the new hyperplane unless $\mathrm{mid}(R_2) = f_2$ (which does not happen).

5. $\mathbf{e}^{1,3} = (0, -1, f_1) \cdot s + (0, 1, f_1)$ never intersects the hyperplane.

And this concludes the proof. $\qquad\square$

We now prove Theorem 8 even for $d > 3$.
**Proof of Theorem 8.** We modify the proof of Theorem 19 to $d > 3$ by adding dummy variables. These dummy variables are denoted by $x_{4:d}$. Furthermore, we add dummy constraints $x_i \geq 0$ for all the dummy variables. We modify the objective function in the proof of Theorem 19 to be $\mathbf{c} = (0, 0, 1, -1, \ldots, -1)$. This will cause all the newly added variables to have no effect on the optimization (they should be set to 0 in the optimal solution) and, hence, the result from Theorem 19 extends to the case when $d > 3$. $\qquad\square$

## B.4    Section 3.4

**Proof of Lemma 10.** First, since all of the points $x_1, \ldots, x_t$ are drawn i.i.d. from $\mathcal{D}$, we observe by symmetry that the event we are interested in is distributed identically to the following event: draw a set of $t$ points $X' = \{x_1, \ldots, x_t\}$ i.i.d. from $\mathcal{D}$ and select an index $i \in \{1, \ldots, t\}$ uniformly at random and compute the probability that $x_i \notin \mathrm{Conv}(X' \setminus \{x_i\})$. In other words

$$\Pr_{x_1,\ldots,x_t \sim \mathcal{D}}[x_t \notin \mathrm{Conv}(X)] = \Pr_{x_1,\ldots,x_t \sim \mathcal{D}, i \sim \{1,\ldots,t\}}[x_i \notin \mathrm{Conv}(X' \setminus \{x_i\})]. \tag{4}$$

We analyze the quantity on the right hand side of (4) instead, fixing the choices of $x_1, \ldots, x_t$, and analyzing the probability only over the randomness of the choice of index $i$. For each edge $e \in E_{\mathcal{P}}$, let $X'_e = X' \cap e$. Since each edge lies on a one dimensional subspace, there are at most two extreme points $x_1^e, x_2^e \in X'_e$ that lie outside of the convex hull of other points i.e. such that $x_1^e \notin \mathrm{Conv}(X' \setminus \{x_1^e\})$ and $x_2^e \notin \mathrm{Conv}(X' \setminus \{x_2^e\})$. We note that when we choose an index $i$ uniformly at random, the probability that we select a point $x \in X'_e$ is exactly $|X'_e|/t$, and conditioned on selecting a point $x \in X'_e$, the probability that $x$ is an extreme point (i.e. $x \in \{x_1^e, x_2^e\}$) is at most $2/|X'_e|$. Hence, we can calculate

$$\Pr\left[x_i \notin \mathrm{Conv}(X' \setminus \{x_i\})\right] = \sum_{e \in E_{\mathcal{P}}} \Pr[x_i \in X'_e] \cdot \Pr[x_i \notin \mathrm{Conv}(X'_e \setminus \{x_i\}) \mid x_i \in X'_e]$$

$$\leq \sum_{e \in E_{\mathcal{P}}} \frac{|X'_e|}{t} \cdot \frac{2}{|X'_e|} = \sum_{e \in E_{\mathcal{P}}} \frac{2}{t} = \frac{2|E_{\mathcal{P}}|}{t}.$$

$\qquad\square$

**Proof of Theorem 9.** First, we show that `LearnHull` makes a mistake only if the true optimal point $\mathbf{x}^{(t)}$ lies outside of the convex hull $\mathcal{C}^{(t-1)}$ formed by the previous observed optimal points $\{\mathbf{x}^{(1)}, \ldots, \mathbf{x}^{(t-1)}\}$. Suppose that at day $t$, the algorithm predicts the point $\hat{\mathbf{x}}^{(t)}$ instead of the optimal point $\mathbf{x}^{(t)}$. Since each point in $\mathcal{C}^{(t-1)}$ is feasible and $\hat{\mathbf{x}}^{(t)}$ is the point with the highest objective value among the points in $\{\mathbf{x} \in \mathcal{C}^{(t-1)} \mid \mathbf{p}' \cdot \mathbf{x} \leq b'\}$, then it must be that $\mathbf{x}^{(t)} \notin \mathcal{C}^{(t-1)}$ because

otherwise $\mathbf{c} \cdot \mathbf{x}^{(t)} > \mathbf{c} \cdot \hat{\mathbf{x}}^{(t)}$. By Lemma 10, we also know that the probability that $\mathbf{x}^{(t)}$ lies outside of $\mathcal{C}^{(t-1)}$ is no more than $2|E_\mathcal{P}|/t$ in expectation, which also upper bounds the probability of LearnHull making a mistake at day $t$. Therefore, the expected number of mistakes made by LearnHull over $T$ days is bounded by the sum of probabilities of making a mistake in each day which is $\sum_{t=1}^{T} 2|E_\mathcal{P}|/t = O(|E_\mathcal{P}| \log(T))$. □

**Proof of Theorem 11.** The deterministic procedure runs $\lceil 18 \log(1/\delta) \rceil$ independent instances of the LearnHull each using independently drawn examples. The independent instances are aggregated into a single prediction rule by predicting using the *modal* prediction (if one exists), and otherwise predicting arbitrarily. Hence, the aggregate prediction is correct whenever at least half of the instances of LearnHull are correct.

We show that if each instance of the LearnHull is run for $8|E_\mathcal{P}|$ days, then the probability that more than half of the instances of LearnHull make a mistake on a newly drawn constraint at day $T + 1$ is at most $\delta$. The result is that with probability at least $1 - \delta$ the majority of instances of LearnHull predict the correct optimal point, and hence the aggregate prediction is also correct.

Let $Z_i$ be the random variable that denotes the probability that the $i$th instance of the LearnHull algorithm makes a mistake on a fresh example, after it has been trained for $8|E_\mathcal{P}|$ days. By Theorem 9, we know $E[Z_i] \leq 1/4$ for all $i$. Now by Markov's inequality,

$$\Pr\left[Z_i \geq \frac{3}{4}\right] = \Pr\left[Z_i \geq 3 \cdot E[Z_i]\right] \leq 1/3,$$

for all $i$. Hence, the *expected* number of instances that make a mistake is at most $1/3$. Finally, since each instance is trained on independent examples, a Chernoff bound implies that the probability that at least half of the instances of LearnHull make a mistake is bounded by $\delta$.

□

## C  Circumventing the Lower Bound when $d \leq 2$

In Theorem 8 (in Section 3.3), we proved the necessity of Assumption 3 by showing that the dependence on the precision parameter $N$ in our mistake bound is tight. However, Theorem 8 requires the dimension $d$ to be at least 3.

We now show that this condition on the dimension is indeed necessary—even without the finite precision assumption (Assumption 3), we can have (computationally efficient) algorithms with small mistake bounds when the dimension $d \leq 2$.

In $d = 1$, at most two constraints are sufficient to determine any constraint matrix $A$ because the constraint matrix $A$ defines a feasible interval on the real line. So we will guess the value that maximizes the objective subject to the single known constraint. Once we have made a mistake, we must have learned the true optimal to the underlying problem because our guess was infeasible. After this single mistake, we either guess the true optimal that we have already seen or if it is not feasible with the new constraint then we guess the point that maximizes the objective subject to the new constraint. Thus after one mistake, the learner will not make any more mistakes.

Lemma 3 tells us that the line between any three collinear points must give us an edge-space of the underlying polytope. When $d = 2$, the corresponding edge-space is then just one of the original constraints of the underlying polytope. Since each solution must be on an edge of the underlying polytope each day, we can make at most $3m$ mistakes without seeing the true objective. Hence, all together, we can make at most $3m + 1$ mistakes before we recover all the constraints of the underlying polytope, or all the rows of the constraint matrix $A$, and see the true optimal solution.

This phenomenon does not continue to hold for $d > 2$ (as we show in our lower bound in Theorem 8).

## D  Missing Proofs from Section 4

First we state Theorem 20 from Grotschel et al. [11] about the running time of the Ellipsoid algorithm.

**Theorem 20.** *Let $\mathcal{P} \subset \mathbb{R}^d$ be a polytope given as the intersection of linear constraints, each specified with $N$ bits of precision. Given access to a separation oracle which can return, for each candidate*

*solution $p \notin \mathcal{P}$ a hyperplane with $N$ bits of precision that separates $p$ from $\mathcal{P}$, the Ellipsoid algorithm outputs a point $p' \in \mathcal{P}$ or outputs $\mathcal{P}$ is empty at most $\mathrm{poly}(d, N)$ iterations.*

We are now ready to bound the number of mistakes that `LearnEllipsoid` makes.

**Proof of Theorem 13.** Whenever `LearnEllipsoid` makes a mistake, there exists a separating hyperplane

$$\mathcal{H}^{(t)} = \left\{ W = (\mathbf{w}^1, \ldots, \mathbf{w}^n) \in \mathbb{R}^{n \times d} \mid \sum_{i \in S^{(t)}} \mathbf{w}^i \cdot (\mathbf{x}^{(t)} - \hat{\mathbf{x}}^{(t)}) > 0 \right\}$$

that cause the Ellipsoid algorithm to run for another iteration. When `LearnEllipsoid` reduces $\mathcal{F}$ to a set that contains a single point up to $N$ bits of precision then predicting via the above equation for $\mathcal{H}^{(t)}$ will ensure we never make a mistake again. Hence, the number of mistakes that `LearnEllipsoid` can commit against an adversary is bounded by the maximum number of iterations for which the Ellipsoid algorithm can be made to run, in the worst case.

We know that $\mathbf{x}^{(t)}$ each day is on a vertex of the polytope $\mathcal{P}^{(t)}$ which is guaranteed to have coordinates specified with at most $N$ bits of precision by Assumption 7. Theorem 12 guarantees that the solution $\hat{\mathbf{x}}^{(t)}$ that `LearnEllipsoid` produces using the following equation

$$\hat{\mathbf{x}}^{(t)} \in \underset{\mathbf{x} \in \mathcal{P}^{(t)}}{\mathrm{argmax}} \left\{ \sum_{i \in S^{(t)}} (\mathbf{w}^i)^{(t)} \cdot \mathbf{x} \right\}$$

is a vertex solution of $\mathcal{P}^{(t)}$. So $\hat{\mathbf{x}}^{(t)}$ can also be written with $N$ bits of precision by Assumption 7. Thus, every constraint in $\mathcal{F}$ and hence each separating hyperplane can be written with $d \cdot N$ bits of precision. By Assumption 6 and Theorem 20, we know that the Ellipsoid algorithm will find a point in the feasible region $\mathcal{F}$ after at most $\mathrm{poly}(n, d, N)$ many iterations which is the same as the number of mistakes of `LearnEllipsoid`. $\qquad\square$