[Reviews · NeurIPS 2016]

Reviewer 1

Summary

This paper studies the problem of sequential prediction of optimal solutions to linear programs with two forms of uncertainty: First, a known objective with unknown feasible set, given by a fixed (unknown) polytope with an additional changing (but known) linear constraint; and second, a changing and unknown utility function with a changing (but known) feasible set. The main results of the paper provide finite mistake bounds on these problems: For the first one, the bound scales linearly with the number of edges and the encoding length of the vertices of the polytope; and the second is shown to be polynomial on the size of the set generating the objectives, the dimension, and the encoding length of the vertices of every feasible set. The paper also provides a lower bound showing that linear dependency on encoding length is tight for the first problem, and stronger results for a PAC version of it. The paper has strong theoretical results, certainly at the level of NIPS. However, the model itself lacks of motivation: It is unclear whether the mistake model is the right way to study this question, and the dependency on the number of edges (which is not shown to be tight) leads to weak bounds in general.

Qualitative Assessment

My overall impression is that this paper has results of high technical quality. However, there are two aspects that I find unsatisfactory: 1. The focus on mistake bounds. This model is very restrictive, and it is unclear it leads to strong consequences beyond what is proved. It is mentioned that mistake bounds imply PAC learnability, but it seems to be too restrictive, which is probably reflected on the obtained bounds. It would be a great addition to further motivate the focus of the paper on mistake bounds. 2. The dependence of the algorithms for the first problem on the number of edges is also restrictive. Beyond the simplex, I don't know other classes of linear programs with nontrivial bounds on the number of edges. On the other hand, no tightness on this parameter is shown, so it is not yet clear this dependence is necessary. The paper would be considerable strengthened if this dependence is shown to be tight.

Confidence in this Review

2-Confident (read it all; understood it all reasonably well)


Reviewer 2

Summary

This paper considers two problems in which the goal is to predict a solution to a partially unknown LP that changes in every round while minimizing the number of mistakes. This setting is inspired by game-theoretic/economic settings in which the agent can observe the results of previous interactions. There are two specific settings considered in this paper. 1. Known objective with fixed unknown constraints and one changing known constraint 2. Unknown objective with known changing constraints. The unknown objective is assume to be a sum of some known subset of fixed objectives. For the first setting two algorithms are given: one for the adversarial setting and one for the stochastic one. Both algorithms are based on reconstruction the edges of the polytope defined by unknown fixed constraints. The second problem is solved via a clever application of an ellipsoid algorithm.

Qualitative Assessment

The questions asked in this paper are novel and I found the solutions to be quite interesting and clever. The paper is well written. The main issue is that there is no underlying motivation that could validate the basic model or various assumptions. So the results are neat but there is also high risk of spending significant effort on a model that does not exist in practice or is not robust to inevitable noise and imprecision. Therefore the work might end up being neither practically relevant nor useful in other theoretical contexts. For example, consider the assumption that the exact solution is given to the player and is unique. Even if all other aspects of the model were valid, in most realistic contexts the constraints might not be known exactly, solutions only close to optimum and not unique. The authors dismiss this issue as making the problem not well-defined and consequently propose a rather fragile-looking solution that relies strongly on collinearity or solutions and reconstruction of the edge space. In summary, I think that the paper has significant theoretical novelty and some potential for applications. But, even by the relatively low standards of theoretical work, it sorely lacks a clear motivating problem that could be used to validate the modeling choices and algorithmic solutions.

Confidence in this Review

2-Confident (read it all; understood it all reasonably well)


Reviewer 3

Summary

The paper provides learning algorithm for predicting the solution of linear program with partial observation and provides some mistake bounds for these algorithms. Two different problems are considered: 1) The objective function of LP is known but the constraint sets are not known. At each time a new constraint is revealed and the goal is to predict the optimal solution of the LP including all known and unknown constraints. 2) The objective function is unknown (it can be a linear combination of different subsets of a collection of fixed but unknown vectors), but the learner observes the changing constraints at each day and again the goal is to predict the optimal solution of the LP. In the first problem, the mistake bounds are obtained for two cases: when the new constraints are selected adversarially, in which case the authors show that given a uniform bound on the precision of the constraints and some other assumptions such as non-degeneracy of the constraints, there exists an algorithm with mistake bound and running time linear in terms of the number of edges of the underlying unknown polytope and the number of precision bits. It is shown that uniform upper bound on the precision bits is essential for dimensions higher than 2. To relax this issue, the authors consider the known objective LP problem in stochastic scenario where the revealed constraints at each time are independently chosen from an underlying unknown distribution, in which case a learning algorithm by expanding convex hulls is provided which attains in expectation at most linear mistakes in terms of the number of edges and grows only logarithmic in time. Moreover, this algorithm has been derandomized to give a deterministic algorithm with high probability bounds. In the case of unknown setting, the authors use a variant of ellipsoid method to design a learning algorithm with polynomial bounds on running time and number of mistakes in terms of parameters of the problem (dimension, precision bits, etc).

Qualitative Assessment

The proposed bounds scale linearly with the number of edges of the unknown polytope. As is mentioned, only when the number of edges is within an additive constant of the dimension, the results in this paper give polynomial bounds in terms of the dimension of the space, and finding an efficient algorithm with polynomial mistakes in terms of dimension is left as an open question. Could the authors please comment on this issue a bit further that what do they think about the existence of such algorithm based on their observations? Also, line 55 talks about halving technique with some performance guarantee. Is there any specific reference for this result? Could you please explain why theorem 18 does not work for d=1, or d=2? The idea of designing learnhull algorithm and its analysis is very interesting. Is there any similar approach which has been used earlier for a similar type of problem? The definition of set \mathcal{F} and the intuition behind it in line 355 is not very clear. Could the authors please clarify it a bit further and explain how it is related to [5]?

Confidence in this Review

2-Confident (read it all; understood it all reasonably well)


Reviewer 4

Summary

This paper studies the problem of "learning from revealed preferences". In this problem, a buyer with linear utility function over a set of objects observes the posted prices on those objects and chooses a set to purchase that maximizes her utility subject to a budget constraint. The learner does not know some part of the optimization problem that the buyer is solving. The goal of the learner is to accurately predict the bundle the buyer will purchase at each time step t. In particular, in this paper the authors consider two different setups: first, the objective is known, set of non-budget constraints is unknown and the posted prices and budget vary over time. In the second setup, objective is unknown (in a specific manner) and the constraint polytope P varies over time. In each setup the goal is to minimize the mistake bound - the number of times the guess does not agree with reality - this assumes adversarial changes in the constraints/objective over time. or the PAC style setting where the constraints change according to some distribution and the goal is to minimize the expected number of mistakes. In the first setting under some reasonable assumptions the authors provide an algorithm with a reasonable mistake bound and running time. This bound assumes that the unknown polytope can be expressed using finite precision N and the mistake bound depends on this parameter N. The authors show that the dependence on the parameter N is necessary. Finally authors provide a result in the PAC setting. For the second setting, authors provide a polynomial time algorithm with polynomial time mistake bound. The approach for this is motivated by the ellipsoid algorithm - they use the optimal solution obtained at each time step to narrow down the space in which the unknown parameters must lie.

Qualitative Assessment

The paper is very well written. It uses non-trivial ideas and insights to derive the results. The results are fairly general so will apply to a broad array of settings. Typo: In section 3.1 first line should say number of mistakes in "LearnEdge".

Confidence in this Review

2-Confident (read it all; understood it all reasonably well)


Reviewer 5

Summary

This paper studies solving linear programming when only partial information of the entire problem is revealed. Two problems, the "known objective problem" and the "known constraints problem", are studied under online setting. The known objective problem involves a fixed known objective function, a fixed set of unknown constraints and a single known constraint changing over rounds. The "LearnEdge" algorithm is proposed with finite mistake guarantee under the assumption of finite precision for vertices of unknown constraints. And the "LearnHull" algorithm is proposed with \log(T) mistake bound under the stochastic scenario. The known constraints problem involves a changing unknown objective function and a set of observable changing constraints. Utilising the ellipsoid algorithm, the problem is solved efficiently with finite mistake, also under the assumption of finite precision.

Qualitative Assessment

+ The paper deals with the very interesting problem of linear programming with partial information, which is the generalisation of the problem known as "learning from revealed preferences". To my opinion, this direction of research is of great practical importance. And the settings considered in the paper are novel and original. The algorithms are proposed with sound theoretical guarantees. - > The major concern is on the assumption of finite precision, which is crucial for proving the mistake bounds. But in practice, this assumption is hard to achieve even when the coefficients in LP are integers. > The definition of the mistake bound (in Definition 1) is somehow too restrictive for the problem, and potentially leads to the difficulty in relaxation of the finite precision requirements.

Confidence in this Review

2-Confident (read it all; understood it all reasonably well)


Reviewer 6

Summary

This paper considers the problem of learning an agent's preferences and constraints given only information about their behavior. Formally, there is a sequence of linear programs with unknown objective/constraints (typically related to each other in some way) and for each program we observe only the optimum (we also potentially have some additional side information, e.g. one of the objective or constraints might be known). The goal is to obtain mistake bounds on predictions of the agent's behavior. The paper has several results: 1. The paper first considers the case where the objective is fixed and known, but there is a (fixed) unknown constraint set, and in each round the constraint set is intersected with a half-space to yield the constraint set for that round. The paper presents an algorithm called LearnEdge which achieves a mistake bound of O(|E|*N*log(d)), where |E| is the number of edges of the polytope, N is the number of bits of precision, and d is the dimension. They then show that the dependence on N is necessary in the online setting. 2. They then turn to the stochastic setting, and show that the bound can be improved to |E|*log(T), where T is the number of rounds. This involves a different algorithm called LearnHull. 3. The authors then turn to the case where the constraints are known but changing, and the objective is unknown. In this case, they present an algorithm called LearnEllipsoid that achieves polynomial sample complexity.

Qualitative Assessment

I think that the paper is well-organized and well-presented, and the results are relatively thorough and interesting. If the paper has a weakness, it would be along the motivational aspect: it wasn't entirely clear to me how important the unknown constraints setting is. However, I think that there is a chance that it turns out to be important, and this is slightly outside my area so I also think that there could already be applications that I'm not aware of. In either case, I think that the paper is suitable for publication at NIPS. The authors should consider explaining the motivation more clearly in the final version, especially in targeting the motivation towards a NIPS audience --- this is the sort of paper that more commonly appears in a venue like EC, though I'm glad that more papers like this one are starting to show up at NIPS and I hope that this trend continues. A few questions I had for the authors: -The mistake bounds in Theorem 1 and Theorem 9 are not directly comparable, due to the dependence on T in the latter bound. If one is willing to incur this log(T) factor in the online setting, is it possible to circumvent the lower bound in Theorem 8, or does it still apply? -Theorem 8 shows that dependence on N is necessary, but what about dependence on |E|? Do you think there are algorithms that don't depend on the number of edges but instead on e.g. the dimension? -I think the number of edges can sometimes be very large even if d is small (e.g. a hypercube). What concrete settings should I have in mind where |E| is well-behaved?

Confidence in this Review

2-Confident (read it all; understood it all reasonably well)